# Predicting PY motif-mediated protein-protein interactions in the Nedd4 family of ubiquitin ligases

**A. Katherine Hatstat, Michael D. Pupi, Dewey G. McCafferty** *

Department of Chemistry, Duke University, Durham, North Carolina, United States of America

* dewey.mccafferty@duke.edu

## Abstract

The Nedd4 family contains several structurally related but functionally distinct HECT-type ubiquitin ligases. The members of the Nedd4 family are known to recognize substrates through their multiple WW domains, which recognize PY motifs (PPxY, LPxY) or phospho-threonine or phospho-serine residues. To better understand protein interactor recognition mechanisms across the Nedd4 family, we report the development and implementation of a python-based tool, PxYFinder, to identify PY motifs in the primary sequences of previously identified interactors of Nedd4 and related ligases. Using PxYFinder, we find that, on average, half of Nedd4 family interactions are likely PY-motif mediated. Further, we find that PPxY motifs are more prevalent than LPxY motifs and are more likely to occur in proline-rich regions and that PPxY regions are more disordered on average relative to LPxY-containing regions. Informed by consensus sequences for PY motifs across the Nedd4 interactome, we rationally designed a focused peptide library and employed a computational screen, revealing sequence- and biomolecular interaction-dependent determinants of WW-domain/PY-motif interactions. Cumulatively, our efforts provide a new bioinformatic tool and expand our understanding of sequence and structural factors that contribute to PY-motif mediated interactor recognition across the Nedd4 family.

## Introduction

Neuronal precursor cell-expressed developmentally downregulated 4 (Nedd4) is the founding member of a family of HECT-type E3 ubiquitin ligases that share a common architecture but have distinct cellular functions. The Nedd4 family is characterized by a multi-domain architecture comprised, from N- to C-terminus respectively, of a C2 domain for membrane localization, two to four WW domains for substrate recognition, and a catalytic HECT domain (Fig 1A) [1–5]. As the final enzyme in the ubiquitin signaling cascade, the Nedd4 family of HECT-type E3 ubiquitin ligases receives ubiquitin from a ubiquitin-E2 conjugating enzyme thioester adduct. The ubiquitin-HECT E3 conjugate then passes ubiquitin to a substrate protein via isopeptide bond formation at target lysine residues. Nedd4 and related HECT-type ligases are thus responsible for conferring substrate specificity in the ubiquitin signaling pathway.

**Funding:** This work was kindly supported by Duke University, National Institutes of Health National

Institute of Neurological Disorders and Stroke Grant 1R21NS112927-01 to D.G.M. (https://www.ninds.nih.gov), Michael J. Fox Foundation Grant 16250 to D.G.M. (https://www.michaeljfox.org), and National Science Foundation Graduate Research Fellowship GRFP 2017248946 to A.K.H. (https://www.nsfgrfp.org/). The funders had no role in study design, data collection and analysis, decision to publish, or preparation of the manuscript.

**Competing interests:** The authors have declared that no competing interests exist.

**Fig 1. The Nedd4 family of ligases contains conserved WW domains for interactor recognition. (A)** Nedd4 and related ligases contain 2–4 WW domains that recognize interactors containing a PY motif (PPxY, LPxY) or phosphorylated threonine or serine residues. **(B)** Alignment of the four WW domains from prototypical member Nedd4 shows moderate sequence similarity and highlights conserved residues, including the two characteristic tryptophan residues (indicated by red arrows). **(C)** Solution state NMR structure of the Nedd4 WW domain 3 (grey) in

complex with a PY motif peptide (red) from a known Nedd4 substrate (PDB ID: 2KPZ, unpublished) reveals key residues (blue) involved in peptide binding.

Understanding the specificity of the Nedd4 family of ubiquitin ligases is of particular interest due to the role of Nedd4 in the regulation of proteostasis in various conditions including cancers [6–8] and neurodegenerative disorders [9–16] and with recent insights into the potential of Nedd4 to serve as a therapeutic target [17–25].

It has been established that the Nedd4 ligases recognize substrates primarily through their WW domains, small structural domains characterized by a three-strand, anti-parallel β-sheet with two conserved tryptophan residues ~20 amino acids apart (Fig 1B) [8, 26–31]. WW domains are found in a variety of proteins and bind primarily to proline-rich regions of target proteins. In the Nedd4 family, WW domain-mediated interactor recognition occurs via binding of the WW domain to a substrate PY motif (PPxY or LPxY, where x can be any amino acid; Fig 1C), or phospho-threonine or phospho-serine residues (pT and pS, respectively) [26–28, 30–36]. There have been extensive efforts to characterize the nature of Nedd4 family ligases and their interactors, from solution state NMR [26, 30, 33, 37–40] and x-ray crystallography [29] characterization of WW domain-PY motif complexes and studies of WW domain-PY motif affinities [41], to affinity-based pull-down assays [42] and high-throughput microarray studies of Nedd4 binders. Through these efforts, significant information about the interactions of Nedd4 family has become available with ~100 to over 700 interactors annotated for different members of the ligase family (Table 1). Comparative analysis of the annotated interactors across the Nedd4 family indicates little overlap between the interactomes of the ligases, indicating that the members of the Nedd4 family are functionally distinct despite high structural conservation (Fig 2A). Gene ontology annotation [43, 44] of the interactomes furthers this, revealing that there are similar trends amongst affected biological processes but distinct patterns in protein classes that interact with each Nedd4 type ligase (Fig 2B).

Using this available interactome data, we sought to analyze the features defining Nedd4 substrate specificity and the PY-dependent substrates of the ligase family. Through this effort, we aimed to characterize the prevalence of canonical PY motifs (both PPxY and LPxY) in the known Nedd4 family interactome to determine the frequency of PY-mediated interactor recognition. Further, we sought to determine the preferred amino acid identity at the x position and the sequence context of the PPxY and LPxY motifs to provide insight into the nature of the protein regions where these domains occur. To this end, we developed a python-based tool, termed PxYFinder, for rapid sequence-based analysis of the Nedd4 family interactome. Analysis of the primary sequences of known Nedd4 family interactors using PxYFinder

**Table 1. Number of previously identified interactors for each Nedd4 family ligase in the BioGrid protein-protein interaction database.**

| Ligase | # of annotated interactors |
|--------|----------------------------|
| Nedd4 | 348 |
| ITCH | 233 |
| SMURF1 | 425 |
| SMURF2 | 150 |
| WWP1 | 112 |
| WWP2 | 763 |
| HECW1 | 26 |
| HECW2 | 308 |

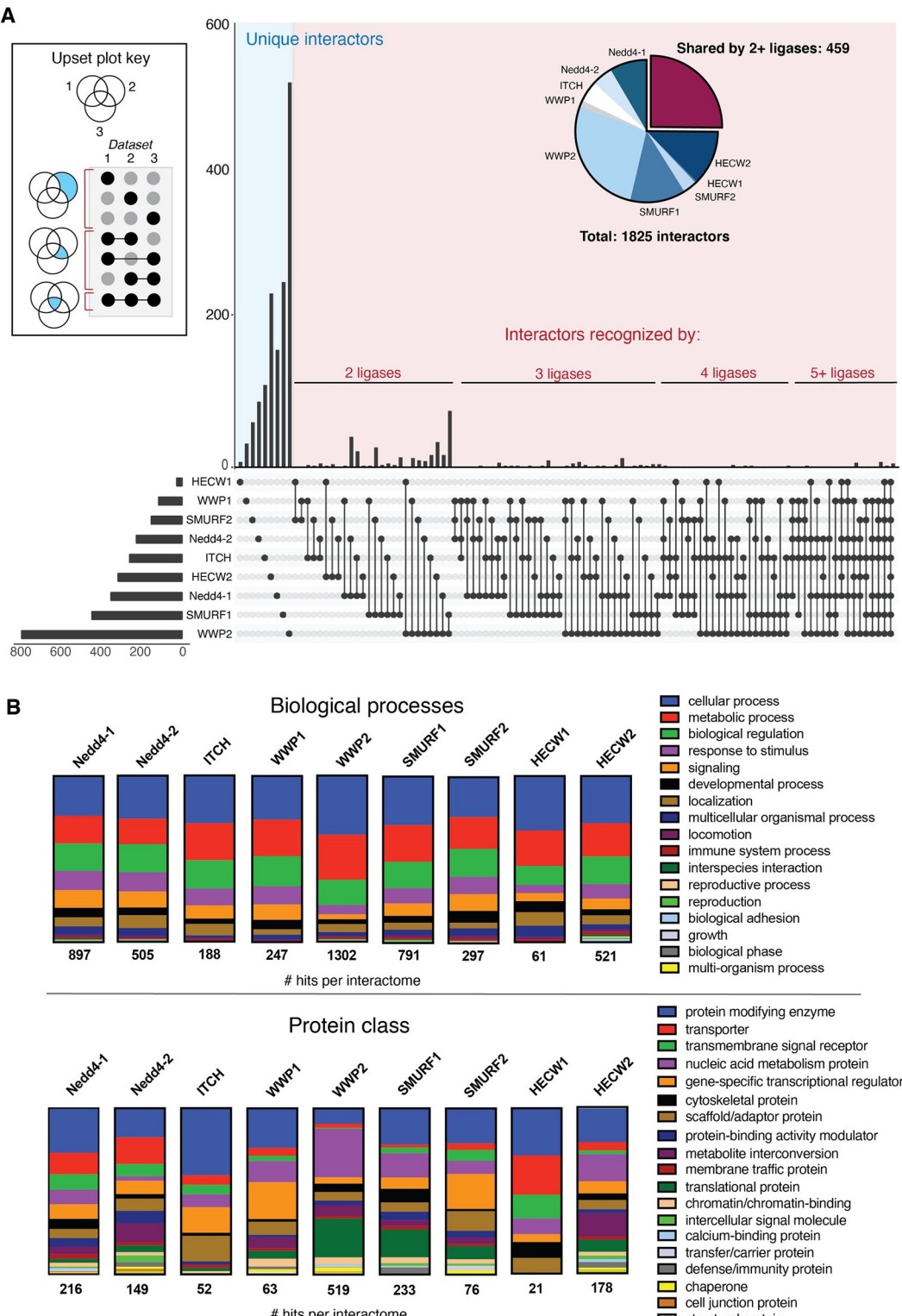

**Fig 2. UpSet analysis and Gene Ontology annotation reveal that Nedd4 family ligases are functionally distinct. (A)** Cross-reference of annotated interactors in the BioGrid database reveals that approximately one quarter of all interactors of the Nedd4 family are recognized by 2+ ligases, revealing little overlap in the known interactomes of the Nedd4 family. Data analysis performed with the UpSet plot tool [76] and graphic annotated in Adobe Illustrator. **(B)** Gene ontology analysis via the PANTHER database [43, 44] reveals that each Nedd4 family ligase interactome has similar trends in biological process (top) but distinct patterns in protein class composition (bottom).

revealed that PY-motifs occur in ~50% of the Nedd4 family interactome, with PPxY motifs occurring more frequently than LPxY motifs. Further bioinformatic analysis reveals that PPxY motifs are both more likely to occur in disordered and in solvent accessible regions than LPxY regions. Next, using consensus sequence data from the PY motif-containing interactors of the Nedd4 interactome, we conducted a computational analysis of PY peptide affinity using a rationally designed peptide library. Specifically, we screened combinations of the most common residues at the $x_{-1}$ and x position (where $x_{-1}$ denotes the residue immediately preceding PPxY or LPxY) using a combination of template-based peptide docking [45] and molecular mechanics-based binding affinity prediction [46] to identify residue-dependent trends in peptide binding affinity. Finally, to gain insight into PY-independent Nedd4/substrate interactions, we conducted an analysis of the non-PY motif containing Nedd4 substrates to identify possible alternative modes of interaction with the ligase. To this end, we screened non-PY substrates against the PhosphoSite database [47] to identify phospho-proteoforms that may drive Nedd4 recognition. Cumulatively, these analyses provide insight into the predominance and nature of PY-motif dependent protein-protein interactions versus PY-independent interactions in the Nedd4 family interactome and establishes a platform for further experimental interrogation of interaction specificity and affinity in the Nedd4 family of ubiquitin ligases.

## Results

### Identification and analysis PY motif sequences in the Nedd4 family interactome

To begin our analysis of PY motif-mediated interactions in the Nedd4 family, we first sought to determine the prevalence of PY motifs amongst interactors of the family. To this end, we retrieved interactome data for the Nedd4 family ligases (Nedd4-1, Nedd4-2, ITCH, WWP1, WWP2, SMURF1, SMURF2, HECW1, HECW2) from BioGrid [48, 49] using *Homo sapiens* as an organismal filter. Across the BioGrid database, the reported interactors have been identified via various means (affinity capture mass spectrometry, affinity capture western blotting, two hybrid, co-localization, biochemical activity assays, etc.). It is important to note that these techniques may also result in identification of indirect interactors (i.e. proteins identified through association with a multi-protein complex), and this idea is discussed further below. Across the Nedd4 ligase family, there are ~100 to over 700 annotated interactors for each of the ligases (Table 1; Fig 2), so we sought a rapid method to screen the interactor sequences for the presence or absence of PY motifs. Since PY motifs can be identified from the protein primary sequence and do not rely on predicted or annotated protein secondary structure or conformation, we developed a python-based script for rapid analysis of protein primary sequence to identify PY motifs (S1A Fig). This script, referred to as PxYFinder, was first validated by analyzing a published dataset in which PY motifs were annotated amongst a pool of proteins. To this end, we chose to employ a dataset by Persaud and co-workers [34] in which Nedd4 interactors were characterized via proteome array and PY motifs in identified Nedd4 interactors were annotated. This dataset included binding partners and ubiquitinated substrates of human Nedd4-1 and Nedd4-2 as well as rat Nedd4-1. Using the published dataset, available as S1 Table (Persaud et al., *Mol Syst Biol*, **2009**, S1 Table at DOI: 10.1038/msb.2009.85) [34], we compiled a list of all PY-motif containing binding partners or substrates identified in screens of all three Nedd4 forms (human Nedd4-1, human Nedd4-2, and rat Nedd4-1). This list was subsequently analyzed with PxYFinder, which revealed that 69 of the 82 reported PY-containing proteins were identified as PY containing proteins with PxYFinder (S1B Fig, S1 Table). Of the identified interactors screened, 13 proteins were annotated as PY motif containing proteins by Persaud and co-workers but were not identified as PY-containing in our analysis (S1B Fig).

To understand this discrepancy, the 13 proteins (canonical and all isoforms) were analyzed manually, revealing that the 13 proteins did not contain the PY motifs that they were reported to contain in the Persaud dataset (S2 Table) [34]. We further manually confirmed that the reported PY motifs were present in the PY-containing proteins that PxYFinder identified (S1 Table). This revealed that all hits (PY containing) and non-hits (no PY motif) as determined with PxYFinder were correctly classified. With this validation, we feel confident that our tool can rapidly identify PY motifs across primary sequences of a large set of proteins.

Using PxYFinder, we identified the prevalence of PPxY and LPxY motifs in the previously interactors of the Nedd4 family ligases as annotated in the BioGrid database (Fig 3A). To this end, we found that that, on average 33.3% of Nedd4 family interactors contain PPxY motifs while 15.7% contain LPxY and 51.0% contain neither PPxY or LPxY motifs. PY motifs appear more prevalent in the Nedd4 family interactomes relative to the annotated *Homo sapiens* proteome. Our analysis revealed that for all ligases studied, PPxY motifs were more prevalent in the interactomes than LPxY motifs. Interestingly, Nedd4-1, Nedd4-2, and ITCH showed similar trends wherein their interactomes were distributed with approximately 40% containing PPxY motifs, 20% containing LPxY motifs, and 40% containing no canonical PY motif. WWP1 showed similar trends with its distribution skewed slightly toward PPxY motif prevalence. For WWP2, SMURF1, SMURF2, HECW1 and HECW2, however, 50% or greater of the interactome lack canonical PY motifs. These results show that on average, approximately 50% of the known Nedd4 family interactome contains a canonical PY motif, providing a sequence-based evidence of the likely mode of interaction between Nedd4 and these substrates.

To further understand PY-dependent interactor recognition in the Nedd4 family interactome, we sought to characterize sequences of identified PY motifs to determine 1) if there was conservation of amino acid identity at the x position in the motif and 2) if there are characteristic features of the protein sequences up and downstream of the PY motif. To this end, we used PxYFinder to extract a slice of the FASTA string of each PY motif-containing interactor that included the identified motif and the 10 amino acids before and after the PY motif. After collecting these extracted sequences across the interactome, we looked for consensus sequences using the WebLogo tool [50] (Fig 3B, S2 Fig). Here, we see moderate conservation of residue identity at the x position of the PPxY motif across the Nedd4 family, with all but SMURF2, HECW1 and HECW2 having proline, serine, and glycine as the three highest probability amino acids in the x position in the consensus sequences. For LPxY containing proteins, the highest probability residue for the x position is shown to be serine or proline for all members of the Nedd4 family except for SMURF1, HECW1 and HECW2. Interestingly, the WebLogo analysis indicates that PPxY motifs are more likely to occur in proline rich regions of the substrate protein than LPxY motifs. In fact, proline is the highest probability residue at almost all of the up and downstream positions for PPxY motifs in substrates of Nedd4–1, ITCH, WWP1, WWP2, SMURF2 and HECW2. On the other hand, there is little sequence consensus up- and downstream of LPxY across the Nedd4 family, with a distribution of charged, polar, and non-polar residues present as highest probability residues across the consensus sequences.

To better understand the propensity of PY motifs to occur in proline-rich regions, we next sought to determine if proline-rich regions are enriched for PPxY motifs relative to chance. To this end, we queried the UniProt database for reviewed *Homo sapiens* protein sequences with annotated compositional bias for proline residues and analyzed these proteins using PxYFinder. This analysis revealed that, in a random sample of ~1300 proteins with annotated proline-rich regions, 16.8% contain PPxY motifs and 11.3% contain LPxY motifs. This analysis reveals that PY motifs (both PPxY and LPxY) occur at lower rates in proline rich regions in general compared in the Nedd4 family interactome (average prevalence in the Nedd4 family was

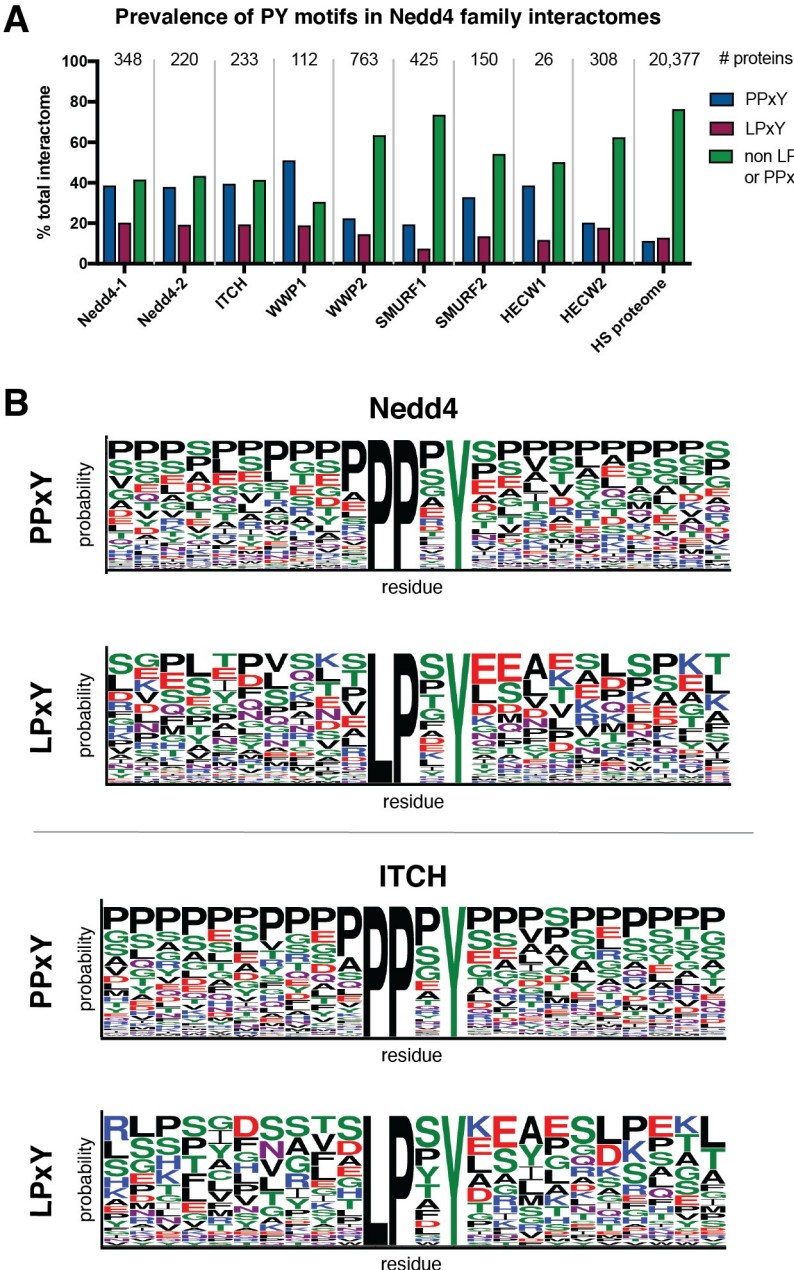

**Fig 3. Analysis of PY motifs in interactomes of the Nedd4 family of ubiquitin ligases. (A)** Prevalence of PPxY and LPxY motifs in the interactome (from BioGrid database) [48, 49] across the Nedd4 family of ubiquitin ligases. **(B)** Representative WebLogo depictions of PY motif consensus sequences from all PPxY and LPxY motifs ± 10 amino acids for Nedd4 and ITCH. The WebLogo [50] diagrams for the remainder of the Nedd4 family ligases are shown in S2 Fig.

found to 33.3% and 15.7% for PPxY and LPxY, respectively). The prevalence of PPxY in these proline-rich regions is higher than in the annotated *Homo sapiens* proteome, where 11.7% of proteins contained PPxY sequences (Fig 3A).

To gain further insight into the structural context of PY motifs in the Nedd4 family interactomes, we sought to characterize the accessibility of PY motifs and the likelihood that these

motifs occur in ordered or disordered protein regions. To this end, a series of bioinformatic algorithms were employed to analyze the relative solvent accessible surface area (RSA) [51], disorder [52], and polyproline helix propensity [53] of the PY motifs identified across the Nedd4 interactome. As a case study, these analyses were conducted with the PY motifs identified in the interactomes of Nedd4-1, WWP2, and SMURF1 as these members of the Nedd4 family have the largest annotated interactome datasets (Fig 2A) and show distinct patterns in the types of protein classes with which they interact (Fig 2B).

For a ligase to recognize an interacting protein through a WW domain/PY motif interaction, it is necessary for the PY motif to be accessible for binding to the WW domain. As a measure of relative accessibility of PY motifs, we sought to characterize the RSA of the PY motifs across the Nedd4-1, WWP2 and SMURF1 interactomes. To this end, bioinformatic tool Net-SurfP-2.0 was employed [51]. This platform employs a deep learning-based algorithm for high throughput prediction of various protein features from primary sequences including propensity for secondary structure formation (helix, coil, or sheet) and solvent accessible surface area. For this analysis, the full primary sequence of each PY-containing interactor was used as we anticipated that analysis of the extracted PY sequence alone may provide insufficient sequence and structural context. Interactors were further delineated by PPxY containing versus LPxY containing. As a measure of accessibility, RSA is calculated as a ratio of the total accessible surface area (ASA) of each residue in a protein structure to the maximum possible solvent accessible surface area of that residue [51, 54, 55]. Calculated RSA, therefore, is value between 0 and 1 where a larger value indicates a higher degree of relative solvent accessibility of a residue or protein region and a lower value indicates a less accessible or more "buried" residue. The Net-SurfP-2.0 algorithm delineates accessible from buried residues with a threshold RSA value of 0.25 [51], such that a residue with RSA < 0.25 is considered buried in the core of the protein and therefore would not be accessible for participation in protein-protein interactions.

Calculation of the predicted RSA of PY motifs across the Nedd4-1 interactome revealed that identified LPxY motifs are more buried than PPxY motifs (Fig 4), with greater than 25% of LPxY motifs having average RSA values less than 0.25 across the motif. Over half of the LPxY motifs have calculated RSA values between 0.25 and 0.5, and just under 25% of the LPxY motifs have high accessibility scores (RSA > 0.5). In contrast, a majority of PPxY motifs have calculated RSA values over 0.5, indicating that a majority of PPxY motifs have higher degrees of solvent accessibility compared to LPxY motifs. This trend is consistent across the motifs analyzed in interactors of SMURF1 and WWP2. This result provides insight into the availability of PY motifs to engage in WW domain mediated interactions and indicates that the presence alone of a PY motif in the primary sequence does not guarantee that the PY motif is accessible for substrate recognition. Therefore, it is important to consider the context of secondary structure in complement with the primary sequence of putative WW domain substrates.

To further explore sequence context, we next employed the IUPred2A [52] tool to calculate the relative order of interactor sequences containing PPxY and LPxY motifs in Nedd4-1, WWP2, and SMURF1. Comparison of the average relative order (wherein a score > 0.5 indicates disorder) of these sequences reveals that, on average across the Nedd4-1 interactome, PPxY motifs occur in more disordered regions that LPxY motifs (Fig 5A). In the case of WWP2 and SMURF1, this trend was retained (S3A Fig). This result is also consistent with the observed trends in RSA in PPxY containing proteins relative to LPxY, where PPxY motifs are on average more solvent accessible than LPxY motifs. We anticipate that this may be a result of PPxY motifs occurring in more proline-rich regions than LPxY motifs (Fig 3B; S2 Fig) as proline-rich regions are associated with intrinsic disorder due to the geometric constraints imposed by the backbone of proline residues [56, 57]. To explore this further, we then

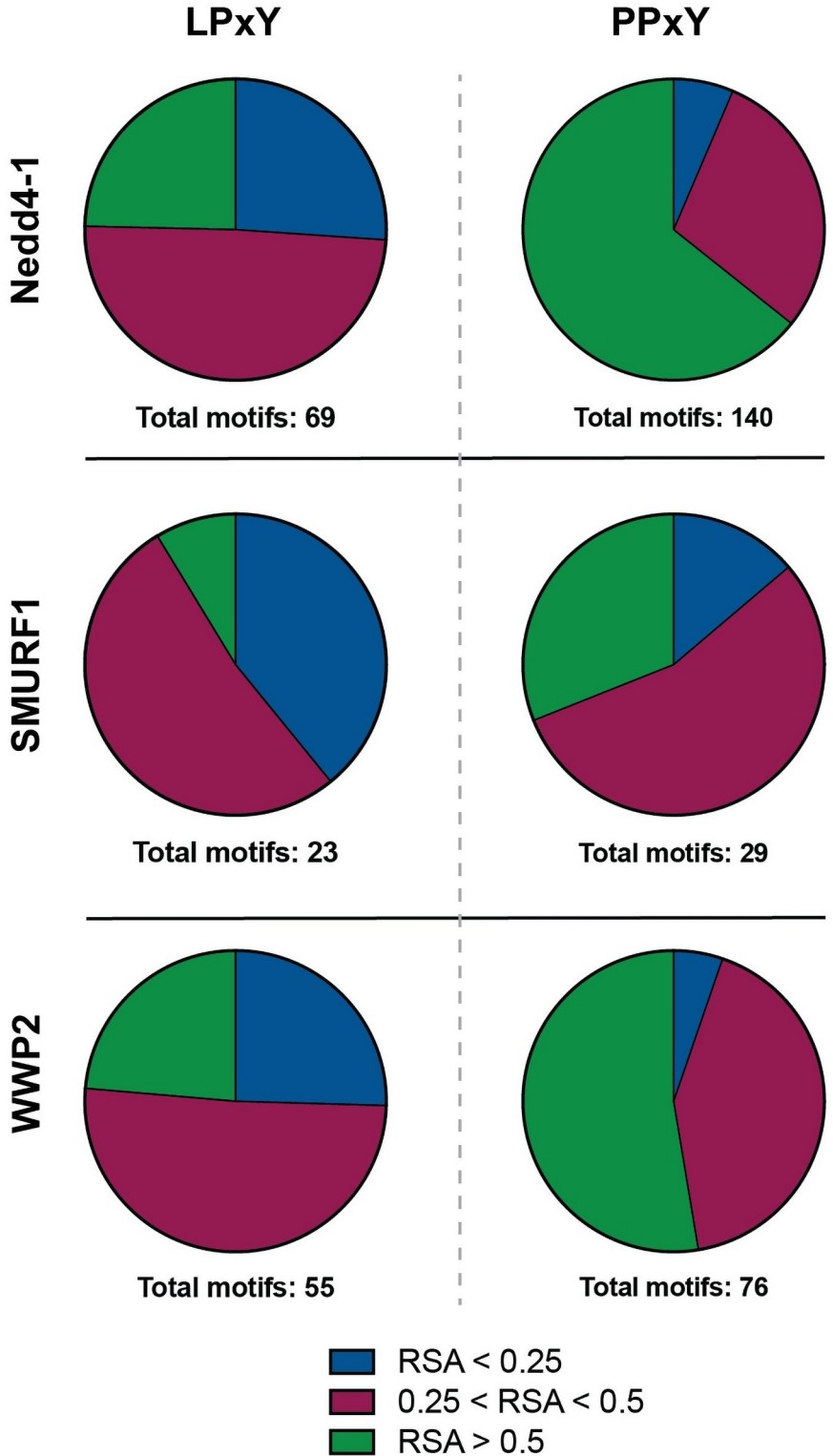

**Fig 4. Relative solvent accessible surface area (RSA) of PY motifs from the Nedd4-1, SMURF1, and WWP2 interactome reveals that PPxY motifs are more solvent accessible than LPxY motifs.** RSA, calculated with NetSurfP-2.0 bioinformatic algorithm [51], is determined by the ratio of total accessible surface area of a residue in the protein relative to maximum accessible surface area of the residue itself. A score of 0.25 or lower is indicative of "buried" residues, or those that would be inaccessible for engaging in protein-protein interactions. Buried residues

(RSA < 0.25) are indicated in blue, moderately accessible (0.25 < RSA < 0.5) in red, and highly accessible (RSA > 0.5) in green. Data visualized with Prism GraphPad.

computationally flipped the identity of the first residue in the PY motif (i.e. P in PPxY substituted with L to afford LPxY) and re-analyzed the sequence disorder using IUPred2A (S3B Fig). In this control study, we observed that the difference in observed disorder is decreased upon substitution of proline for leucine or vice versa. This result indicates that, while the PPxY motifs appear to occur in more disordered regions, the presence of the first proline residue in the motif is likely a large contributor to that disorder.

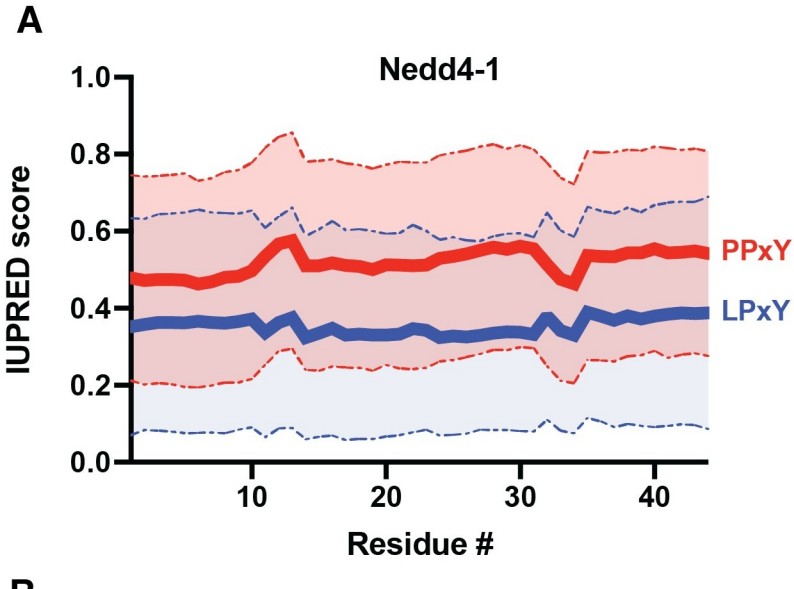

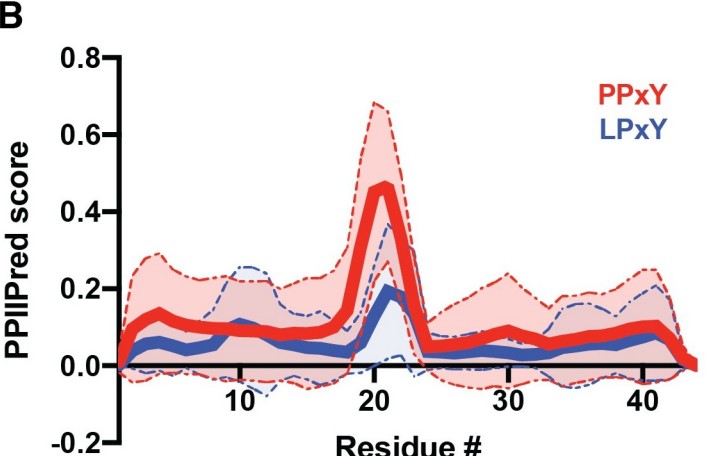

**Fig 5. Prediction of relative order of PY-motif containing regions from the Nedd4 interactome.** PY motif sequences ± 20 amino acids were extracted from the Nedd4 interactome using PxYFinder script and analyzed using **(A)** IUPred2A [52] and **(B)** PPIIPred [53] bioinformatic tools to determine relative order and propensity to form polyproline II structure, respectively. Data shown as average ± S.D. of 109 (PPxY) and 41 (LPxY) sequences. Statistical analysis using a paired t-test (to compare at each residue, numbered 1–44 above) reveals a statistically significant difference in predicted order between the PPxY and LPxY sequences (p < 0.0001 for both IUPred and PPIIPred scores). Analysis across the sequence using an unpaired Welch's t-test also shows significant differences (p < 0.0001 for IUPred; p < 0.002 for PPIIPred).

We then sought to analyze the effect of proline prevalence on the predicted order of the PY motif-containing regions via PPIIPred [53], a bioinformatic tool for identification of polyproline II (PPII) secondary structure, an extended helix-like structure that can occur in the presence of polyproline stretches. PPIIPred analysis reveals that PPxY motifs are more likely to display PPII structure immediately before or at the PY motif (residues 20–24 in extracted sequence slices) as compared to LPxY motifs (Fig 5B). The sequences up and downstream of the PY motifs, however, show similarly low propensities for PPII structure on average. We anticipate that the increased prevalence of proline residues in PPxY-containing regions contributes to relative disorder but does not induce PPII structure on average.

## Rational design of PY motif peptide library for computational analysis

Analysis of previously resolved PY motif/WW domain complex structures from Nedd4 show moderate conservation of PY peptide backbone conformation regardless of primary sequence (Fig 6A). To better understand the effect of PY sequence on WW domain binding, we sought to determine if the sequence variants of the PY motif affected the predicted affinity with which the substrate of interest binds to Nedd4. To this end, we began with a previously resolved structure of a Nedd4 WW domain bound to a PY-motif peptide from a known substrate, sodium channel ENaC (PDB ID: 2M3O) as our model complex [30]. Using the ENaC peptide (sequence: TAP**P-PAY**ATLG, with PY motif in bold) as a template, we designed a peptide library based on the previously described consensus sequences. We chose to vary the residues at the x and $x_{-1}$ position of the PY motifs (where $x_{-1}$ is the residue immediately preceding PPxY or LPxY) as these are the residues which span the binding interface between the PY peptide and WW domain (Fig 6B). Based on the consensus sequences (Fig 3B, S2 Fig), we generated 15 variants each for PPxY and LPxY peptides using the template peptide (TA**x$_{-1}$PPxY**ATLG or TA**x$_{-1}$LPxY**ATLG) with all combinations of the three and five highest probability residues at the $x_{-1}$ and x positions, respectively (Fig 6C). It should be noted that, as there is not a previously characterized complex of a hNedd4 WW domain bound to a PY-peptide with the LPxY motif, we opted to use the same template peptide for screening of both PPxY and LPxY motifs to allow for direct comparison across the suite without variation outside of the peptide core. Cumulatively, this design afforded 30 peptides in total for computational screening against a Nedd4 WW domain.

## Docking and molecular mechanics analysis of WW domain/PY motif interactions

Prior to computational analysis of our rationally designed peptide library, we sought to determine the WW domain scope required to capture any sequence-dependent variation in WW domain binding across the Nedd4 family. To this end, we compared the conservation of WW domain sequences across the family of ligases. Each ligase contains 2–4 WW domains (Fig 1A), with moderate sequence similarity across the family (Fig 1B **and** S4A and S4B Fig). Analysis of key residues that interact with peptide substrates shows moderate conservation of the binding interface (S4C Fig). These residues, which are primarily located in the concave peptide binding cleft of the three-stranded β-sheet structure, drive the direct interaction of the WW domain with the PY motif (S4C Fig). Alignment of three representative WW domain structures with varied residue identity in the binding interface (Nedd4–1 (WW3) and ITCH (WW3 and WW4), all of which have been shown to bind ENaC, the substrate from which the peptide library was derived) shows conservation of secondary structure (S4C Fig). Additionally, analysis with MolProbity [58] and KiNG [59] indicate that the interactions between the peptide substrate and WW domain are predominantly mediated by van der Waals contacts with few

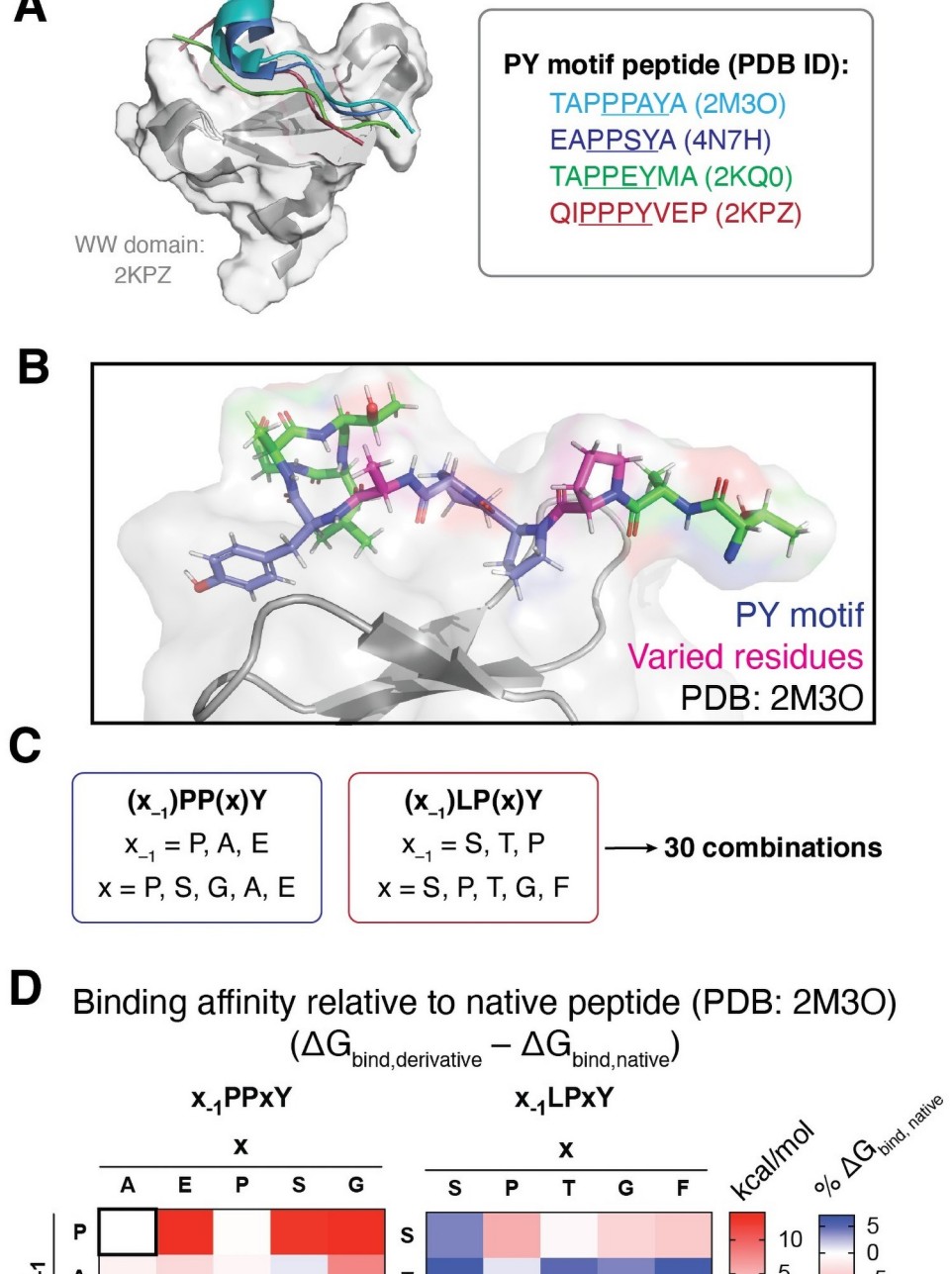

**Fig 6. Rational design and computational analysis of PY motif peptide library to predict residue-specific changes in binding affinity. (A)** Nedd4 WW domain (PDB ID: 2KPZ) in complex with PY motif peptides from previously resolved WW domain/PY peptide complexes (PDB IDs: 2KPZ, 2M3O, 2KQ0, 4N7H). Peptides aligned to the 2KPZ complex using PyMol [75], showing moderate conservation of peptide backbone conformation when bound to the WW domain. **(B,C)** Rational design of PY peptide library involved variation of residues in the $x_{-1}$ and x positions (shown in pink, **B**) and was informed by PY motif consensus sequences for Nedd4 (shown in Fig 2C and S2 Fig), affording a 30 member library. **(D)** Computationally predicted binding affinities of PY motif peptides screened against Nedd4 WW domain (PDB ID: 2M3O). Binding affinities are presented as $\Delta\Delta G_{binding}$ relative to the native peptide substrate TAPPPAYATLG ($\Delta G_{binding}^{designed} - \Delta G_{binding}^{native}$). $\Delta\Delta G_{binding}$ for the native peptide is presented in the upper-right corner of the left heatmap for reference. $\Delta\Delta G_{binding}$ energies presented as kcal/mol or % of $\Delta G_{binding}^{native}$. Full energy properties provided as in S1 Data.

inter-peptide and peptide-WW domain hydrogen bonds. Therefore, we anticipate that trends in binding affinity across the peptide library from screening against our model WW domain (PDB: 2M3O) will be representative as the electrostatic nature of the binding interface is largely conserved. Instead, we anticipate that sequence-dependent changes in the peptide-protein interactions and peptide conformation will have a greater effect on binding affinity than the identity of WW domain residues.

To begin our computational analysis of the rationally designed PY motif library against our suite of WW domain structures, we first sought a docking method that was amenable to docking peptides to protein targets rather than small molecule ligands. We determined that use of a template-based docking method was the most appropriate approach for our analysis as there are a number of PY peptide/WW domain complexes that have been reported. Therefore, known PY/WW complex structures can be used to guide the docking, improving efficiency and minimizing computational expense compared to a global docking approach. To this end, we employed GalaxyPepDock [45] to dock our library against a Nedd4 WW domain (PDB ID: 2M3O) [30]. Each GalaxyPepDock docking result provided 10 predicted poses for the peptide of interest. From these 10 poses, we selected the pose with the most similarity to the native substrate in the PY/WW complex (PDB ID: 2M3O) (S5 Fig). To further refine the docking result, we then used the Glide ligand docking tool with SP-Peptide precision setting from the Schrödinger suite [60–62] to optimize the conformation of the peptide backbone and side chains in the binding pocket. Finally, to obtain thermodynamic measurement of predicted peptide binding affinities, we employed molecular mechanics-based binding affinity prediction using the Generalized Born and surface area continuum solvation method (MM/GBSA, Schrödinger) [46], which considers the effect of solvation on binding energies using an implicit solvation model. From this calculation, we generated a total measurement of affinity as $\Delta G_{binding}$ in addition to various contributing energy terms, enabling analysis of biomolecular interactions that serve as driving forces in the peptide/protein interaction.

The predicted $\Delta G_{binding}$ values provide a relative measure of affinity across the peptide suite wherein a more negative number indicates a stronger predicted peptide/protein interaction. Docking results were analyzed by comparison to the predicted binding affinity of the native peptide (Fig 6D). Our docking analysis reveals that, in general, substitutions at the $x_{-1}$ or x position in the PPxY peptide scaffold weaken the predicted binding affinity (indicated by a less negative $\Delta G_{binding}$) with the exception of derivatives APPSY, EPPPY, and EPPGY. We anticipate that there is a significant deal of pre-organization in the native ligand around the tri-proline core (TA**PPP**AYATLG), and we hypothesize that alteration of the steric or electrostatic nature at the x position with retention of the tri-peptide core (PPPxY) is unfavorable as the peptide lacks flexibility to compensate for altered interactions with the WW domain. Screening of derivatives with alanine or glutamic acid at the $x_{-1}$ position was slightly unfavorable, but derivatives APPSY, EPPPY demonstrated improved affinity, likely through an increased number of intramolecular interactions due to the bent conformation adopted by the optimized docked ligand (S6 Fig).

In the LPxY peptide library, the derivatives generally had stronger predicted binding affinities than the PPxY library members. We anticipate that this is a result of greater ligand flexibility resulting from the lessened conformational strain induced by the core proline-proline dipeptide. We also anticipate that the greater hydrophobicity of leucine relative to proline may drive binding of the PY peptide to the WW domain pocket, contributing to the trend observed. Several members of this peptide class that have strong predicted binding affinities adopted a bent conformation, increasing the number of intramolecular contacts. Further, we hypothesize that the increased polarity with substitutions of serine or threonine at the $x_{-1}$ or x positions increases either dipole-mediated intramolecular interactions or stabilizes the peptide/WW

domain complex by presenting the polar residue to the solvent accessible side of the peptide and promoting burial of the lipophilic residues in the WW domain binding pocket.

We then analyzed individual energetic contributions to overall binding affinity across the library of peptide analogues. This includes energy components of the free ligand or receptor, the optimized complex, or sub-components of the $\Delta G_{binding}$ measurements (i.e. contributions of individual interaction types). In general, van der Waals and Coulombic interactions contributed most strongly to binding affinity, while solvation energy accounted for the most disfavorable (positive $\Delta G$) component (Fig 7A and S7 Fig). We next correlated all individual energy components to total $\Delta G_{binding}$ (Fig 7B and 7C and S8 Fig). Analysis of energy components from complex, ligand, and receptor showed that receptor energies had the lowest correlation with overall $\Delta G_{binding}$ while ligand and complex energy components had higher correlations (Fig 7B). Further analysis showed that ligand efficiency, a function of binding affinity relative to total non-hydrogen atoms, correlated most strongly with $\Delta G_{binding}$ (Fig 7C). Finally, this analysis reveals that van der Waals are most correlated with $\Delta G_{binding}$, followed by Coloumbic and lipophilic interactions.

## Analysis of non-PY motif substrates in the Nedd4 interactome

While we have extensively discussed the nature of PY motif-mediated interactions with WW domains, the nature of interactions that guide the remaining half of the interactome remain unclear from our analysis. It is likely that these interactions are guided by interactions at other sites in the ubiquitin ligase, such as is the case for E2 conjugating enzymes [63–65], which interact with the HECT domain, or for proteins like α-synuclein [11–14, 16], which has been shown to interact with the C2 domain and HECT domain of the ligase. Additionally, there is evidence for WW domain interactions with phospho-threonine or phospho-serine (pT, pS) [26]. In these cases, the Nedd4 interaction would be dependent upon specific phospho-proteoforms, the presence of which are regulated by other cellular pathways and is discussed further below.

To complement our analysis of PY motif-mediated protein-protein interactions in the Nedd4 family interactome, we sought to further analyze the pool of non-PY motif interactors in the annotated dataset. We first performed a functional analysis of non-PY interactors using the PANTHER GO annotation database [57, 58] to determine how many proteins in the interactome were involved in the ubiquitination process (for example, E2 conjugating enzymes that would bind to the ligases through the E2 interaction site on the catalytic HECT domain). From this annotation, we identified that the non-PY containing interactome contained a range from 2.19% (WWP2) to 11.63% (WWP1) across the Nedd4 family of Nedd4 (Table 2). This indicates that nearly all of the non-PY interactome is not comprised of upstream members of the ubiquitin signaling cascade but rather contains substrates or regulatory partners that interact with Nedd4 in a PY-independent manner (i.e. through phosphorylated residues or through C2, HECT, or linker interactions).

To characterize the remainder of the non-PY containing proteins in the Nedd4 interactome, we next screened non-PY interactors for the presence of pT or pS residues as reported in the PhosphoSite protein phosphorylation database. Of the 153 non-PY interactors identified in the Nedd4 interactome, there are 128 proteins that are annotated in PhosphoSite database [47] to contain both pT and pS post-translational modifications (PTMs) while 17 proteins have been detected with either pT or pS and eight proteins have no reported pT or pS residues. Based on these previous reports, experimentally detected phosphorylation on threonine and/ or serine residues occurs in 94.8% of the non-PY interactome. Thus, this provides putative evidence that phosphorylation at serine or threonine may be the driving force for Nedd4

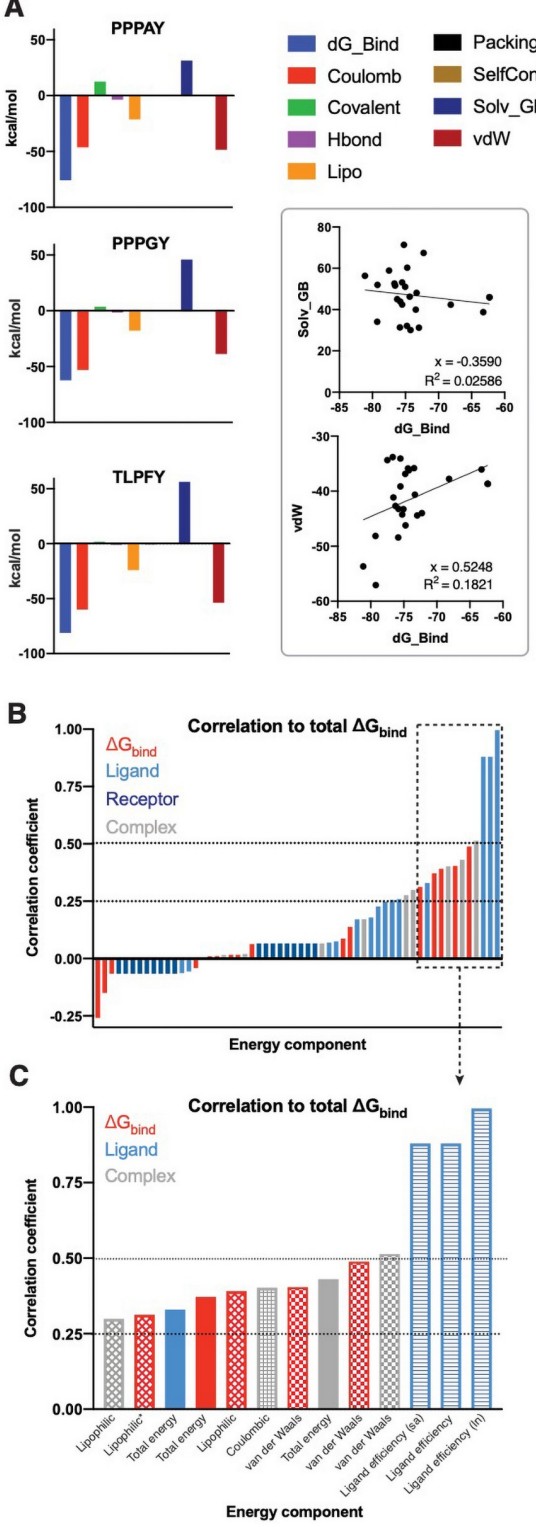

**Fig 7. Individual biomolecular interaction types have varying contributions to overall binding affinity. (A)** Energetic contributions of individual interactions to overall binding affinity are shown for the native ligand (PPPAY), a weaker predicate binder (PPPGY), and a stronger predicted binder (TLPFY). Linear regression analysis reveals a positive and negative correlation, respectively, between van der Waals forces or solvation energy with overall binding energy ($\Delta G_{binding}$). **(B)** Analysis of all individual energy components (for $\Delta G_{bind}$, the optimized complex, ligand, and

receptor) to overall binding affinity reveals factors that ligand and complex energies are more strongly correlated than receptor energies. Ligand efficiency is defined as the binding energy/# heavy atoms where "sa" accounts for solvent exposed surface area and ln is the natural log of ligand efficiency. * indicates lipophilic interactions in the $\Delta G_{bind,NS}$ where NS indicates binding energy of the peptide without accounting for ligand strain energies. Correlations calculated using python as Pearson coefficients and visualized in Prism GraphPad.

recognition of non-PY interactors. Though experimental validation of putative WW domain/phospho-protein interaction specificity would be required, it is beyond the scope of this investigation.

As a final investigation of potential mechanisms underlying protein recognition by the Nedd4 family, we sought to compare the interactome of Nedd4 and those of non-PY, non-pS or pT interactors to determine if there are shared interactors between the identified protein and Nedd4. To this end, annotated protein-protein interactions of a small sampling of non-PY, non-phosphorylated Nedd4 interactors were cross referenced with the Nedd4 interactome (S9 Fig). This analysis revealed a number of shared secondary interactors that contain PY motifs or are known to have pT or pS proteoforms. This evidence indicates a potential role of protein complex formation in the identification of Nedd4 interactors. We hypothesize that the non-PY, non-phosphorylated interactors may be part of a larger protein complex that is recognized by Nedd4 and, thus, these proteins were enriched in affinity capture methods and were subsequently annotated in the Nedd4 interactome. Therefore, their recognition by Nedd4 may not be through direct formation of a protein-protein interaction with Nedd4 but is instead scaffolded by other Nedd4 interactors in a larger complex. There is evidence for this idea of substrate clustering, as demonstrated by Mund and Pelham [66], who determined that Nedd4 more efficiently recognized polymerized or clustered substrates relative to the monomeric or isolated forms.

Finally, it is important to note that Nedd4 has a number of disordered linker regions that may influence the interaction specificity of the enzyme. It is known that the Nedd4 linker domains contribute to autoregulation of Nedd4 activity by forming or facilitating intramolecular interactions [67–69], but increasing knowledge of protein interactions has revealed that disordered linker regions also participate in protein-protein interaction events [70–73]. Therefore, it is possible that non-PY motif interactions may be mediated through binding in disordered regions of the Nedd4 family enzymes.

## Discussion

We have employed a combination of bioinformatic and computational analyses to gain insight into the sequence and structural properties that drive interaction specificity in the Nedd4 interactome. We began our analysis with the development and implementation of PxYFinder to rapidly identify the presence or absence of canonical PY motifs in a library of FASTA

**Table 2. Functional analysis of non-PY containing interactors involved in ubiquitination.**

| Ligase | # non-PY motif interactors | % of non-PY interactome involved in ubiquitination |
|---|---|---|
| Nedd4 | 144 | 3.73 |
| ITCH | 96 | 11.46 |
| SMURF1 | 312 | 5.36 |
| SMURF2 | 81 | 10.94 |
| WWP1 | 34 | 11.63 |
| WWP2 | 483 | 2.19 |

sequences. Using this tool in combination with interactome data available through the BioGrid database, we determined that, on average, 33.8% of Nedd4 family interactors contain PPxY motifs while 15.5% contain LPxY and 50.6% contain neither PPxY or LPxY motifs. This demonstrates that canonical PY motifs drive only half of the WW-domain mediated interactions in the known interactome on average, and that screening for PY motifs is not sufficient on its own for identification of putative Nedd4 family substrates. In general, all members of the Nedd4 family that we analyzed have more interactors that contain PPxY motifs than LPxY motifs, and consensus sequence analysis reveals that PPxY motifs are more likely to occur in proline-rich, disordered, and solvent accessible regions than LPxY motifs. While this analysis expands our understanding of the prevalence of PY motifs in these interactome, it should be noted that the presence of a PY motif sequence in an interactor does not guarantee recognition by a Nedd4 family ligase. For instance, our bioinformatic analysis revealed that some of these sequences may be buried in the protein core and are thus inaccessible for WW domain/PY motif-mediated recognition. Therefore, future efforts may focus on experimental validation of these specific recognition modes across the Nedd4 family. Despite this, the combination of bioinformatic analyses employed reveals the prevalence as well as the sequence and structural context in which these motifs occur.

Using the information obtained from PxYFinder and consensus sequence analysis of the Nedd4 interactome, we then sought to computationally analyze sequence-dependent effects on PY peptide/WW domain binding. To this end, we employed a multi-step computational analysis of peptide binding affinity using a previously resolved structure of the Nedd4 WW domain. Specifically, we designed a library of PY peptides (both PPxY and LPxY motifs) which contain all combinations of the three and five most commonly occurring residues at the $x_{-1}$ and x positions based on our bioinformatics-derived consensus sequences. We then employed a multi-step computational analysis wherein we began with template-based docking of the peptide substrate to the WW domain structure, followed by refinement of the complex and analysis of thermodynamic binding parameters via MM-GBSA. From this effort, we determined that the PPxY scaffold is less tolerant to substitutions than the LPxY scaffold. We hypothesize that this is a result of pre-organization in the poly-proline backbone of the PPxY peptides. Therefore, incorporation of residues that increase peptide flexibility or polarity tend to improve binding affinity. As predicted, our analysis reveals that binding affinity is most strongly driven by van der Waals interactions, with positive though lesser correlations to Coulombic and lipophilic interactions.

To gain further insight into the role of WW-domain binding in Nedd4 family substrate recognition, we analyzed the non-PY containing interactome of Nedd4 as a case study. This analysis reveals that nearly all of the non-PY substrates have been previously annotated to have phosphorylation at threonine and/or serine residues, providing a putative indication of WW-domain recognition independent of canonical PY motifs. While experimental validation of these hypotheses would be necessary to confirm the mechanism of Nedd4 recognition, our bioinformatic analysis provides valuable insight into possible modes of binding.

Cumulatively, the results presented herein provide insight into the prevalence and nature of PY motifs in the Nedd4 interactome. We anticipate that PxYFinder will be useful in screening large datasets for putative WW-domain interactors (both in the Nedd4 family and for other WW domain-containing proteins) and addresses a gap in current bioinformatic tools for which there is not an established method for identification of PY motifs in a large dataset. Further, our analysis of identified PY motifs expanded our understanding of the conservation of residues in and around the motif. Specifically, we demonstrated that, despite differences in interactor specificity that cause the Nedd4 family ligases to be functionally distinct, trends in sequence and structural context of the PY motifs are largely conserved across the family. This

indicates that the specificity is driven not by protein structure alone but to a higher level of regulation. Finally, our efforts informed a computational analysis of sequence-dependent changes in PY peptide binding affinity. While the binding parameters obtained in this computational analysis are relative, we anticipate that our results will be useful in informing experimental design of PY peptide libraries either for interrogating the nature of the peptide/protein interaction or for designing inhibitors that target PY peptide/WW domain complexes.

## Methods

### Development and use of PxYFinder script

A python script, termed PxYFinder, was developed in Python 3.8 to perform the following workflow: PxYFinder imports FASTA sequences and iterates through primary sequences to identify PPY, PPxY, or LPxY. If a PY motif is identified, PxYFinder extracts a slice of the FASTA string that contains the PY motif and x (user-specified) amino acids up and down stream, copying this slice to a new.csv file. Code and documentation for PxYFinder is available as S1 File.

For analysis of Nedd4 family interactors, interactome data for each Nedd4 family member of interest (Nedd4, ITCH, WWP1, WWP2, SMURF1, SMURF2, HECW1, HECW2) was retrieved from BioGrid using *Homo sapiens* as an organismal filter. Gene names were converted to UniprotIDs and were used to retrieve FASTA sequences from the Uniprot database. PxYFinder was used to identify and extract PY motifs in the interactome of each ligase and calculate prevalence of PY motifs in each interactome. Graphs were generated using Prism (GraphPad). PY motif consensus sequences were determined by analysis with WebLogo (http://weblogo.threeplusone.com/) [50] with probability as the y-axis measure.

### Prediction of protein order in PY-motif containing segments

Interactor sequences were subsequently analyzed using the IUPred2A and PPIIPred tools for prediction of overall disorder (IUPred2A) [52] and propensity to form polyproline secondary structures (PPIIPred) [53]. Data was visualized as mean ± S.D. and analyzed using paired and unpaired (Welch's) t-test to determine statistical differences between specific residue positions (numbered 1–44) and across the full sequence, respectively. Data visualization and analysis was performed in Prism (GraphPad).

Relative solvent accessible surface area was calculated across the full primary sequence of identified PY-motif containing proteins using NetSurfP-2.0 [51], and calculated RSA of the four PY motif residues in each protein was extracted and averaged across the motif. Data visualization performed in Prism GraphPad.

### Docking and molecular mechanics analysis of PY peptide library

A 30-member peptide library was generated based on the consensus sequences in the PY motif interactome of Nedd4. The top three and five residues at the $x_{-1}$ and x positions respectively were paired in all possible combinations to generate 15 peptides each for PPxY and for LPxY libraries using a previously characterized substrate peptide bound to a Nedd4 WW domain (PDB ID: 2M3O) [30] as a template. All peptides were first docked via GalaxyPepDock [45] using 2M3O as a template structure. Docked complexes were further refined using the Schrödinger suite (Schrödinger Release 2020–3, Schrödinger, LLC, New York, NY, 2020). Specifically, the complex generated using GalaxyPepDock was prepared using the Protein Preparation Wizard and LigPrep tools [74]. Using the Glide tool [60], a docking grid for the WW domain was generated using (Glide Receptor Grid Generation), and the ligand was

docked as a flexible ligand to the generated grid using SP-Peptide function with retention of amide bond conformation and restriction of docking poses to 0.50 Å tolerance for core pattern comparison relative to the native ligand conformation. Following docking, the Schrödinger Prime MM-GBSA [46, 61, 62] tool was used to analyze the PoseView (PV) docking output file with the VSGB solvation model and OPLSe3 force field to generate $\Delta G_{binding}$ for each generated pose. For each peptide, the pose with most negative $\Delta G_{binding}$ was selected for comparison of predicted binding affinities across the peptide library. Graphs of binding data were generated using Prism GraphPad. Structural analysis and visualization were performed with PyMol [75], MolProbity [58], and/or KiNG [59].

## Supporting information

**S1 Fig. PxYFinder tool enables rapid identification of PY motifs in large sets of protein primary sequences. (A)** The workflow of PxYFinder implements a python-based script to rapidly identify PY motifs from protein sequences as FASTA format. Protein interaction datasets can be retrieved from public databases such as BioGrid. PxYFinder script allows conversion from interaction list to UniProt ID for FASTA accession. FASTA sequences are then processed as data strings for identification of PY motif and extraction of PY-containing regions. **(B)** Validation of PxYFinder script with manual confirmation against a previously published dataset34 of PY motif-containing proteins reveals errors in previously identified PY motifs.
(TIF)

**S2 Fig. Sequence logo analysis reveals trends in PY motif consensus sequences across the Nedd4 family.** Sequence logo diagrams were used to identify consensus sequences in PY motifs and in surrounding regions (± 10 amino acids) for Nedd4 family members and for all *Homo Sapiens* proteins that have SwissProt annotation available in the UniProt database (labeled as HS proteome). Sequence logos for Nedd4-1 and ITCH are excluded from this figure as they are presented as representative images in Fig 2. Sequence logo analysis reveals that PPxY motifs are more likely to occur in proline-rich regions than LPxY motifs, and amino acid identity at the x position is more conserved in PPxY motifs across the Nedd4 family and proteome than in LPxY motifs.
(TIF)

**S3 Fig. Analysis of predicted order reveals similarities in PY-motif containing regions of representative ligase interactors. (A)** As a first analysis, the predicted order of each PY motif containing Nedd4-family interactome member was analyzed using IUPred2A and disorder scores were extracted ± 20 amino acids surrounding the PY motif sequence. Nedd4-1, SMURF1, and WWP2 show similar trends in predicted order around the PY motifs, with PPxY motifs occurring in more disordered regions relative to LPxY. **(B)** PY motifs in each interactor were computationally flipped wherein PPxY was substituted for LPxY and vice versa. Interactors were then re-analyzed with IUPred2A, revealing that substitution of P for L in the PY motif decreased predicted disorder values in PPxY-containing proteins while substation of L for P increased predicted disorder. This trend was consistent for all three interactomes analyzed.
(TIF)

**S4 Fig. Nedd4 family WW domain sequence and structure alignment show moderate sequence and high structural similarity.** Sequence alignments of WW domains from Nedd4 family members sorted by **(A)** ligase and **(B)** similarity show moderate sequence conservation, with high conservation of key residues in the binding interface (highlighted in grey). **(C)** Alignment of three WW domain structures with varying sequence similarity show high

conservation of structure and of positioning of key residues despite differences in residue identity.
(TIF)

**S5 Fig. GalaxyPepDock accurately predicted conformation of substrate peptide based on template.** As a test of GalaxyPepDock template-based docking reliability, the native peptide substrate of Nedd4 WW domain (reported in PDB structure 2M3O) was docked to the apo-WW domain, extracted from PDB 2M3O. Alignment of the native complex (2M3O, peptide shown in red; WW domain in grey) with the docked complex (via GalaxyPepDock; peptide in green; WW domain in blue) show reliable docking of the peptide with retention of conformation and peptide-WW domain contacts.
(TIF)

**S6 Fig. Conformations of selected peptide derivatives after computational docking and optimization.** A sampling of peptide conformations from computational docking of the rationally designed peptide library demonstrates the variety of intramolecular contacts that the PY peptides can form with the WW domain structure. Binding energies of the representative peptides shown here are presented in Fig 4.
(TIF)

**S7 Fig. Energetic contributions to computationally predicted $\Delta G_{binding}$ of PY peptide library to Nedd4 WW domain.** $\Delta G_{binding}$ and energetic components that contribute to $\Delta G_{binding}$ are shown here as calculated with the Schrodinger Prime MM-GBSA tool. Energies are given in kcal/mol, and energy contributions are shown for all 30 members of the rationally designed PY peptide library.
(TIF)

**S8 Fig. Correlation of energetic components that contribute to peptide binding.** Correlation of calculated energies ($\Delta G_{binding}$ and $\Delta G_{binding}$ sub-components) across the peptide library show that **(A)** some energetic contributions are more strongly correlated to overall binding ($\Delta G_{binding}$) relative to other components. **(B)** Correlation of solvation (Solv_GB) and van der Waals (vdW) components of $\Delta G_{binding}$ with coulombic interactions shows that solvation is more strongly correlated with coulombic interactions than van der Waals interactions. Specifically, stronger (more negative) coulombic interactions correlate with more positive solvation energies. Values calculated with Schrodinger Prime MM-GBSA and presented in kcal/mol. Simple linear regression analysis and data visualization performed in Prism GraphPad.
X = slope of linear regression line of best fit; $R^2$ provided as measure of goodness of fit.
(TIF)

**S9 Fig. Protein interaction network analysis reveals shared secondary interactors and functional links of non-PY containing Nedd4 interactors. (A)** Interaction networks of Nedd4-1 (green node) and non-PY, non-pT/pS substrates of Nedd4 (blue nodes) were retrieved from BioGrid and merged using Cytoscape, revealing secondary interactors that are functionally related and contain either PY (red triangles) or pT and/or pS residues (red squares). **(B)** Identity of primary and secondary interactors depicted in **A** are presented where bolded proteins contain pT and/or pS residues while italicized proteins contain PY motifs.
(TIF)

**S1 Table. PY-containing proteins correctly identified from test set using PxYFinder.**
(DOCX)

**S2 Table. Proteins identified as non-PY containing with PxYFinder but labeled as PY containing in test data set (from Persaud et al., 2009) [34].**
(DOCX)

**S1 Data.**
(CSV)

**S1 File.**
(ZIP)

## Acknowledgments

The authors would like to thank the Duke University Department of Chemistry Computing Services for access to the Schrödinger software suite and computational servers. They also thank the department for support for M.D.P. through the summer undergraduate research fellowship program. Finally, the authors thank the members of the McCafferty lab for their thoughtful feedback on the project and manuscript.

## Author Contributions

**Conceptualization:** A. Katherine Hatstat, Dewey G. McCafferty.

**Data curation:** A. Katherine Hatstat, Michael D. Pupi.

**Formal analysis:** A. Katherine Hatstat, Dewey G. McCafferty.

**Funding acquisition:** Dewey G. McCafferty.

**Investigation:** A. Katherine Hatstat, Michael D. Pupi.

**Methodology:** A. Katherine Hatstat.

**Project administration:** Dewey G. McCafferty.

**Software:** A. Katherine Hatstat, Michael D. Pupi.

**Supervision:** Dewey G. McCafferty.

**Validation:** A. Katherine Hatstat.

**Visualization:** A. Katherine Hatstat, Michael D. Pupi.

**Writing – original draft:** A. Katherine Hatstat.

**Writing – review & editing:** A. Katherine Hatstat, Dewey G. McCafferty.

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
