## [Decision Letter · Decision Letter 0]

7 Jul 2021

PONE-D-21-16013

Predicting PY motif-mediated protein-protein interactions among the Nedd4 family of ubiquitin ligases

PLOS ONE

Dear Dr. McCafferty,

Thank you for submitting your manuscript to PLOS ONE. After careful consideration, we feel that it has merit but does not fully meet PLOS ONE’s publication criteria as it currently stands. Therefore, we invite you to submit a revised version of the manuscript that addresses the points raised during the review process.

Please carefully consider the first reviewer comments with regard to comparing observed motif frequencies to random expectation, and the consideration of 'flipped motifs' that maintain amino acid composition but vary sequence.  The second review points out a number of potential limitations to the extensibility of this analysis beyond human Nedd4 that should be addressed in the discussions.  

We look forward to receiving your revised manuscript.

Kind regards,

Vikas Nanda, Ph.D.

Academic Editor

PLOS ONE

2. We note that you have referenced (PDB ID: 2KPZ )which has currently not yet been accepted for publication. Please remove this from your References and amend this to state in the body of your manuscript: (PDB ID: 2KPZ : [Unpublished]”) as detailed online in our guide for authors

Reviewers' comments:

Reviewer's Responses to Questions

**Comments to the Author**

1. Is the manuscript technically sound, and do the data support the conclusions?

Reviewer #1: Yes

Reviewer #2: Partly

2. Has the statistical analysis been performed appropriately and rigorously? 

Reviewer #1: N/A

Reviewer #2: N/A

3. Have the authors made all data underlying the findings in their manuscript fully available?

Reviewer #1: Yes

Reviewer #2: Yes

4. Is the manuscript presented in an intelligible fashion and written in standard English?

Reviewer #1: Yes

Reviewer #2: Yes

5. Review Comments to the Author

Reviewer #1: Review of Haststat et al. for PLOS ONE

This work was a computational analysis of the targets of Nedd4 and related E3 ubiquitin ligases. This family had been described to bind to PPxY or LPxY motifs and to phosphorylated threonine and serine residues. This work nicely quantifies these binding interactions. The authors downloaded known interaction partners and scanned them for motifs or phosphorylated residues. The paper focuses on Nedd4 and it’s binding targets but also does a nice job of generalizing to the other family members.

The paper is very well written and is accessible to a non expert reader. The introduction is particularly clear. The section on pg 5 describing the benchmarking against a published dataset from Persaud et al. 2009 was thorough.

This paper makes three claims:

On Nedd4 family targets PPxY motifs are more prevalent than LPxY motifs

PPxY motifs are more likely to occur in proline rich regions

Regions containing PPxY motifs are more disordered than regions containing LPxY motifs

Claim 1 is well supported by the data. The authors downloaded interaction partners for six Nedd4 family proteins from BioGrid. They developed a python software program, PxYFinder, to identify PPXY and LPxY motifs. In all, 49% of targets contained motifs.

On page 5, the authors state: “The prevalence of PY motifs in the Nedd4 family interactomes is enriched relative to the annotated Homo sapiens proteome.” This statement is not supported by the data presented. How often do these motifs occur by chance in the proteome?

As written, Claims 2 and 3 are supported by the data but there are several controls that could strengthen these claims.

In the abstract, Claim 2 is stated as “PPxY motifs are more likely to occur in proline rich regions.” The data presented in Figure 3 and S2 clearly show that PPxY motifs occur in regions rich in proline.

However, it is not clear if proline-rich regions are enriched for PPxY motifs compared to what would be expected by chance. What is the frequency that PPxY motifs occur by chance in proline rich regions? It would be interesting to download all annotated proline rich regions from Uniprot and ask how frequently these PPxY motifs occur by chance. Or to scramble the sequences of the target proteins and ask how often motifs occur by chance. This analysis is not essential but would strengthen the claim.

Claim 3, that regions containing PPxY motifs are more disordered than regions containing LPxY motifs, is supported by using IUPred2A to predict the disorder of the 44 residues surrounding the motif (20 AA upstream, the motif and 20 AA downstream). It appears that IUPred2A was run on this 44AA sequence in isolation. Earlier editions of IUpred had artifacts at the beginning and end of show sequences (edge effects). It is possible IUPred2A has fixed this problem. For short sequences, IUpred can overestimate disorder. In my group, we run IUpred on the full protein and then extract the values for the residues of interest. I recommend the authors check at least one example to ensure there are no edge effects in the IUPred2A calculation.

There are two analyses that contrast the context of the PPxT and LPxY motifs: solvent accessibility and disorder. Solvent accessibility (RSA) is computed with NetSurfP-2.0. Disorder is computed with IUpred. Both of these algorithms are composition based. There is a potential that the composition difference of the motifs LPxY vs PPxY can bias these computations. P’s promote exposure and disorder while L’s promote inaccessibility and order (Oldfield and Dunker).

Oldfield, C. J. & Dunker, A. K. Intrinsically Disordered Proteins and Intrinsically Disordered Protein Regions. Annual Review of Biochemistry 83, 140307200228009 (2013).

For example, in Figure 5A, I am concerned that some of the difference between the IUPRED scores between the two sets of sequences is due to the composition of the motif. In the IUPRED energy matrix, P’s increase the Disorder score and L’s decrease the disorder score. This result would be strengthened if the authors computationally “flipped the motifs”—replaced L with P and vice versa—repeated the IUPRED calculations and showed the difference remained. This control would show that the difference in predicted disorder is solely due to the surrounding sequence context and not due to the motif composition. This analysis is not necessary, but would allow the authors to separate the role of motif context from the role of motif composition. In a perfect world, the authors would remove the motif and calculate the disorder of the context alone, but that analysis causes other problems.

Similarly, I think the result in Figure 4 would be strengthened by repeating the analysis on “flipped motif” sequences. This analysis would allow the authors to again separate the role of motif context from the role of motif composition.

I really liked the analysis in Figure 5B. I am curious what it would look like with the motifs flipped. Is one instance of “PP” enough to increase the PPIPred score?

The design of the synthetic peptides and peptide docking simulations was very interesting and I enjoyed reading this portion of the paper. I did not feel qualified to rigorously assess this portion of the manuscript. I defer to the judgement of other reviewers for this section.

On pg. 11, it would be helpful to the reader if the authors discussed if and how the hydrophobicity of the L in the LPxY motif might contribute to the stronger binding.

The analysis of the components of deltaG was very interesting. I would appreciate a discussion of how this binding compares to other examples.

On pg 13, the analysis to separate upstream and downstream targets was strong.

Overall this manuscript is a sound piece of original work.

Minor notes:

In Figure 2, in the pie chart, it is not clear what the colors refer to. Can the bar graphs in the lower left panel match the colors in the pie chart? I think the pie chart could be larger. Overall a very nice figure.

pg 7, “For this analysis, the full primary sequence of each PY-containing interactor of Nedd4-1 was used as use of the extracted PY sequence may provide insufficient sequence and structural context.” This sentence was very hard to read. Please consider revising it.

Figure S2. I think it would be nice to quantify the total fraction of residues that are proline in each set.

Supplment pg 7. There might be ‘.’ missing after van der Waals interactions.

Reviewer #2: This manuscript describes a new tool named PxYfinder, which is used to identify PY motif in the sequence of Nedd4 family proteins’ interactors. And it analyzes the nature of PPxY/LPxY motif-containing proteins and non-PY motif-containing proteins. In my opinion, the main issues with this manuscript are as following:

1、The authors mainly discussed the differences among PPxY/LPxY containing and non-containing Nedd4 family interacting proteins, but the interactors of the Nedd4 family were compiled from BioGRID database only using Homo sapiens as an organismal filter. The interactions of BioGRID were collected from multiple methods like Affinity CaptureMS, Affinity Capture-Western, and so on. The authors should discuss the reliability of the Nedd4 family proteins' interactors.

2、Proteins containing PY motif but not interact with Nedd4 family proteins (PY motif non-substrate proteins) should also be discussed.

3、According to the authors' analysis in Figure 2, Nedd4 family proteins rarely share interacting proteins, but the authors only selected Nedd4 as a study case to investigate the RSA and PPII secondary structure. The natures of Nedd4’s interactors can’t be generalized to other Nedd4 family proteins’ interactors.

4、Furthermore, the authors should also discuss the nature’s differences among different Nedd4 family proteins' substrates.

Minor points:

1、Perhaps it is more appropriate to express Nedd4-1 and Nedd4-2 proteins as NEDD4 and NED4L，to distinct the Nedd4 family and Nedd4 protein.

2、The legend of Figure 6D can’t be found. And I suspect the heatmap legend wasn’t colored.

3、References 37 and 38 seem to be the same.

6. PLOS authors have the option to publish the peer review history of their article (what does this mean?). If published, this will include your full peer review and any attached files.

Reviewer #1: **Yes: **Max Staller

Reviewer #2: No

---

## [Author Response · Author response to Decision Letter 0]

20 Aug 2021

Response to Reviewers

Reviewer 1:

Reviewer 1 commented on the claim that “PPxY motifs are more likely to occur in proline rich regions,” noting that it is not clear if proline-rich regions are enriched for PPxY motifs compared to what would be expected by chance. To address this, we conducted the suggested analysis wherein proteins with reported compositional bias for proline residues were retrieved from UniProt and analyzed with PxYFinder. This analysis revealed that PY motifs occur with greater prevalence in the Nedd4 family interactomes than they do in the proline rich regions queried. A discussion of this analysis is provided on Page 7. 

Reviewer 1 further provided thoughtful feedback about the occurrence of end-effects in IUPred2A and noted a possible issue with our implementation of the tool to analyze extracted PY sequences instead of full interactor sequences. To address this issue, we first re-analyzed the full Nedd4-1 interactome with complete protein Fasta sequences and compared this to the results garnered with extracted PY sequences. We observed that use of the extracted sequences did in fact result in a slightly higher estimation of disorder relative to full sequences, but nonetheless the overall conclusion of the analysis was the same (i.e. PPxY occur in more disordered regions than LPxY). The IUPred2A data has been updated to include the data from analysis of complete protein Fasta sequences, and further analyses were included on the PY-motif containing interactors of SMURF1 and WWP1 as part of the response to Reviewer 2 concerns addressed above. Figure 5 and methods have been also updated accordingly. 

On a related note, we also conducted Reviewer 1’s recommended control study where we computationally flipped PPxY motifs for LPxY motifs and vice versa. In this analysis, provided in a new supplemental figure (Figure S3), we determined that swapping P for L decreased the predicted disorder and L for P increased the predicted disorder, indicating that the first P residue in the motif contributes greatly to the overall estimation of disorder in the region. This trend was retained across the ligase interactomes studied. In addition to the supplemental figure, a brief discussion of this experiment was added on Page 9. This was a straight-forward but informative control experiment, and we greatly appreciate the suggestion! 

Finally, Reviewer 1 recommended that a brief discussion of hydrophobicity of L vs P were added on Page 11. We added a brief discussion of this as requested. 

Reviewer 2:

Reviewer 2 noted that the interacting proteins were retrieved from BioGrid with Homo sapiens as an organismal filter and detailed several of the interaction identification methods that are compiled in the database. Based on this, the reviewer requested that we discuss the reliability of these data and any bias that may occur in the datasets based on the types of methods employed. To this end, we provided a short discussion of methods used to experimentally identify interacting proteins in the Results section on Page 4. 

Reviewer 2 requested that we incorporate a discussion of the possibility that proteins may contain a PY motif but not interact with Nedd4 family proteins. A short discussion of this idea was added to the Discussion section on Page 16.

The reviewer further requested that we expand the discussion of difference among the Nedd4 family substrates. To address these concerns, several revisions were made including:

1. The UpSet plot analysis in Figure 2 was complemented with a functional annotation of known interactors of each ligase using Gene Ontology terms via the PANTHER database. This additional analysis furthered the idea that each ligase is functionally distinct. Specifically, it revealed that, while there are similar trends in the overall biological processes affected by each ligase, there are distinct patterns in protein class amongst the interactomes across the Nedd4 family. This data was added to Figure 2 as Figure 2B. 

2. In the initial submission, the subsequent bioinformatic analyses focused on Nedd4-1 only. To account for the lack of overlap across the family (and therefore the lack of generalizability), we chose to expand these analyses to also include WWP2 and SMURF1. As we have now explained on page 7, the analyses now include these three ligases “as these members of the Nedd4 family have the largest annotated interactome datasets (Figure 2A) and show distinct patterns in the types of protein classes with which they interact (Figure 2B).” We feel that this addition allows for analyses across ligases with distinct specificities to see if our findings are more generalizable. However, by limiting to the largest interactome datasets, we limit potential biases that may occur with inclusion of limited datasets such as HECW1, SMURF2, WWP1, which include fewer than 100 interactors. Interestingly, the sequence and structural context (as determined via the included analyses of RSA, disorder, etc.) show consistent trends across the ligases studied. We feel that the conservation of the trends across Nedd4-1, SMURF1, and WWP2 shows that the results are generalizable despite differences in substrate specificity and points to a level of regulation above that of sequence or structure context of PY motifs alone. 

Additionally, the additional minor errors in the manuscript identified by both Reviewers have been corrected in this version of the manuscript as requested:

1. Clarification of the pie chart in Figure 2

2. Revision of a few sentences that were written with confusing or unclear wording

3. Correction of numbering/lettering in Figure 6 caption 

4. Removal of redundant references (37 and 38 in original submission)

---

## [Editor Report · Decision Letter 1]

24 Sep 2021

Predicting PY motif-mediated protein-protein interactions in the Nedd4 family of ubiquitin ligases

PONE-D-21-16013R1

Dear Dr. McCafferty,

We’re pleased to inform you that your manuscript has been judged scientifically suitable for publication and will be formally accepted for publication once it meets all outstanding technical requirements.

Kind regards,

Vikas Nanda, Ph.D.

Academic Editor

PLOS ONE

---

## [Editor Report · Acceptance letter]

1 Oct 2021

PONE-D-21-16013R1 

Predicting PY motif-mediated protein-protein interactions in the Nedd4 family of ubiquitin ligases 

Dear Dr. McCafferty:

I'm pleased to inform you that your manuscript has been deemed suitable for publication in PLOS ONE. Congratulations! Your manuscript is now with our production department. 

Kind regards, 

on behalf of

Dr. Vikas Nanda 

Academic Editor

PLOS ONE